# The Biology of Stress Intolerance in Patients with Chronic Pain—State of the Art and Future Directions

**DOI:** 10.3390/jcm12062245

**Published:** 2023-03-14

**Authors:** Arne Wyns, Jolien Hendrix, Astrid Lahousse, Elke De Bruyne, Jo Nijs, Lode Godderis, Andrea Polli

**Affiliations:** 1Pain in Motion Research Group (PAIN), Department of Physiotherapy, Human Physiology and Anatomy, Faculty of Physical Education and Physiotherapy, Vrije Universiteit Brussel, 1090 Brussels, Belgium; arne.wyns@vub.be (A.W.); astrid.lucie.lahousse@vub.be (A.L.); jo.nijs@vub.be (J.N.); andrea.polli@vub.be (A.P.); 2Department of Public Health and Primary Care, Centre for Environment & Health, KU Leuven, Kapucijnenvoer 35, 3000 Leuven, Belgium; lode.godderis@kuleuven.be; 3Flanders Research Foundation-FWO, 1090 Brussels, Belgium; 4Chronic Pain Rehabilitation, Department of Physical Medicine and Physiotherapy, University Hospital, 1090 Brussels, Belgium; 5Rehabilitation Research (RERE) Research Group, Department of Physiotherapy, Human Physiology and Anatomy, Faculty of Physical Education & Physiotherapy (KIMA), Vrije Universiteit Brussel, 1090 Brussels, Belgium; 6Department of Hematology and Immunology-Myeloma Center Brussels, Vrije Universiteit Brussel, 1090 Brussels, Belgium; elke.de.bruyne@vub.be; 7Unit of Physiotherapy, Department of Health and Rehabilitation, Institute of Neuroscience and Physiology, Sahlgrenska Academy, University of Gothenburg, 405 30 Gothenburg, Sweden; 8External Service for Prevention and Protection at Work, IDEWE, 3001 Heverlee, Belgium

**Keywords:** chronic pain, stress intolerance, autonomic nervous system, hypothalamus-pituitary-adrenal axis, genetics, epigenetics

## Abstract

Stress has been consistently linked to negative impacts on physical and mental health. More specifically, patients with chronic pain experience stress intolerance, which is an exacerbation or occurrence of symptoms in response to any type of stress. The pathophysiological mechanisms underlying this phenomenon remain unsolved. In this state-of-the-art paper, we summarised the role of the autonomic nervous system (ANS) and hypothalamus-pituitary-adrenal (HPA) axis, the two major stress response systems in stress intolerance. We provided insights into such mechanisms based on evidence from clinical studies in both patients with chronic pain, showing dysregulated stress systems, and healthy controls supported by preclinical studies, highlighting the link between these systems and symptoms of stress intolerance. Furthermore, we explored the possible regulating role for (epi)genetic mechanisms influencing the ANS and HPA axis. The link between stress and chronic pain has become an important area of research as it has the potential to inform the development of interventions to improve the quality of life for individuals living with chronic pain. As stress has become a prevalent concern in modern society, understanding the connection between stress, HPA axis, ANS, and chronic health conditions such as chronic pain is crucial to improve public health and well-being.

## 1. Stress Intolerance Plays a Major Role in Chronic Widespread Pain

Chronic pain affects approximately 20% of the global population and is associated with a significant burden for the individual and their significant others [1]. It is moreover influenced by several cognitive, emotional, and social factors [2]. Stress is one such factor that is able to influence pain symptoms and has long been proposed as relevant in the pain experience [3]. The World Health Organization (WHO) defines stress as any type of change that causes physical, emotional, or psychological strain [4]. The stress response is the physiological and biological response of the body to any situation causing such strains [5].

Stress is highly subjective. Different individuals might respond differently to the same stressful situation. The stress response, therefore, depends on the perceived amount of stress as well as on the nature, duration, and intensity of the stress stimulus [6,7,8]. In patients with chronic pain, stress is generally associated with a worsening of pain symptoms and stress-induced hyperalgesia. In fact, stress and pain are highly comorbid, and show significant overlap in both conceptual and biological processes [9]. On the one hand, experiencing stressful events in life puts individuals at risk to develop chronic musculoskeletal pain and patients with symptoms of post-traumatic stress disorder report higher pain severity levels [3,10]. On the other hand, dealing with chronic pain increases the risk to develop stress-related conditions such as depression and anxiety [11]. Furthermore, a recent review showed that a blunted acute stress response predicted chronic pain and poor health at a long-term follow-up (1 year) [12].

However, the impact of stress in patients with chronic pain goes beyond pain modulation. Other symptoms such as fatigue and cognitive symptoms can also be triggered or worsened because of stress [13,14]. Here, we define the exacerbation or occurrence of symptoms in response to stress as stress intolerance.

## 2. Objectives

This state-of-the-art paper aims to provide an overview of the biological mechanisms that may explain stress intolerance in patients with chronic pain, focussing on the two major stress systems—the autonomic nervous system (ANS) and the hypothalamic pituitary axis (HPA). Although stress intolerance can be induced by physical and mental stress, this state-of-the-art paper focuses on evidence originating from studies investigating mental stress.

Of note, other biological systems should not be ignored when aiming to unravel the pathophysiology of stress intolerance in patients with chronic pain. Considering that stress intolerance comprises various symptoms within different domains, it probably stems from a multisystemic pathophysiology. Other systems showing intricate links with the ANS, the HPA, nociceptive mechanisms, and the stress response are thus likely, collaboratively with the ANS and HPA axis, involved in explaining stress intolerance in chronic pain. The immune system, as well as mechanisms related to the opioid and endocannabinoid system, can all potentially influence and be influenced by pain and stress. We acknowledge the complexity of the aforementioned systems and their interactions. However, a detailed description of such systems is beyond the scope of this review and can be found elsewhere [15,16,17,18,19].

## 3. Methodology

A search exploring stress system dysregulations in chronic pain was queried on PubMed and Web of Science up to December 2022 using following keywords such as chronic pain, stress physiology, autonomic nervous system, SAM axis, HPA axis, hyperalgesia, (nor)adrenaline, catecholamine, cortisol, glucocorticoids, stress hormone, stress response, (epi)genetics, immunology. Inclusion criteria for relevant articles were: (1) address one of the scopes within this review; (2) describe a rationale for the state-of-the-art aspect; (3) written in English or Dutch; (4) human studies or animal studies if necessary.

## 4. Two Major Stress Systems: The Autonomic Nervous System and the Hypothala-Mus-Pituitary-Adrenal Axis

The stress response is an evolutionary conserved, complex, and efficient system with modulation in associated neural (CNS), endocrinological, and immunological systems [20]. Perception of a stressor activates several neuronal circuits involving the limbic forebrain, the brainstem, and nuclei of the hypothalamus, which on their part release stress-mediating molecules, initiating a stress response [21]. Physical and psychological stressors activate different neural networks, resulting in a specified stress response [20]. Physiological and behavioural mechanisms simultaneously aim to restore body homeostasis and promote stress adaptation [22]. The two main neural circuits through which our body adapts to stress are the autonomic nervous system (ANS) and the Hypothalamus-Pituitary-Adrenal (HPA) axis (see Figure 1 for a schematic overview). These systems usually work in synchrony and influence each other through mutual, positive feedback loops [23].

Under normal circumstances, acute physical or psychological stressors activate the ANS inducing a short-lasting increase in sympathetic nervous system (SNS) activity. Stress activates brainstem catecholaminergic neurons and efferent spinal cord neurons of the dorsal intermediolateral column, which converge in pre-ganglionic sympathetic neurons [24]. These neurons synapse directly to chromaffin cells in the adrenal medulla, which secretes adrenaline and noradrenaline in the circulation. In addition, other pre-ganglionic neurons project to several post-ganglionic sympathetic neurons in paravertebral ganglia, using acetylcholine (ACh) as neurotransmitter. Consequent activation of nicotinic receptors on these post-ganglionic neurons results in noradrenaline secretion at the target tissue [25]. Adrenaline and noradrenaline have diverse physiological functions, depending on the adrenergic receptor (AR) they bind to. ARs are G-protein-coupled receptors and can be divided in α1-, α2- and α1-, β2-, and β3-ARs. The overall effect of α1- and α2-ARs activation is increased heart rate (HR) and blood pressure (BP), and decreased heart rate variability (HRV). Blood flow is increased to the skeletal muscles and decreased towards the abdominal organs, metabolic activity such as glycogenolysis in skeletal muscle and lipolysis in adipocytes are promoted to increase energy availability [24]. On the contrary, β1- and β2-ARs stimulation foster vasodilation, decrease blood pressure and increase HRV, though can either increase or decrease HR [26,27,28]. Several organs, as well as immune cells, express both α- and β-ARs, allowing fine regulation of their functions. Decreased expression of β-ARs have been associated with several inflammatory conditions such as rheumatic diseases and obesity [29,30]. β2-ARs show potent anti-inflammatory effects [23], and their down-regulation or desensitisation can help explain pain symptoms.

The HPA axis provides a protracted response, yet its activation is delayed compared to the SNS. This response originates when the hypothalamus, the paraventricular nucleus (PVN) in particular, is triggered by stressors. The PVN releases several neurochemicals, such as oxytocin, vasopressin, and corticotrophin-releasing hormone (CRH) [31,32]. CRH reaches the anterior pituitary (adenohypophysis) and stimulates it to synthesise and secrete adrenocorticotrophic hormone (ACTH) [32]. ACTH, on its part, stimulates the cortex of the adrenal gland to produce and release glucocorticoids, mostly cortisol [33]. Cortisol in turn also exerts an effect on the PVN and anterior pituitary, by limiting synaptic plasticity and suppressing neural excitability, thus creating a long and short negative feedback loop [22]. Glucocorticoid secretion in humans follows a general ultradian and circadian rhythm with basal peak cortisol levels around weaking-up time [34]. Cortisol exerts its functions through binding mineralocorticoid receptors (MR) or glucocorticoid receptors (GR), both ligand-activated transcription factors [35]. These receptors are widely expressed throughout the body. Not surprisingly, cortisol affects several organs and systems [36]. The HPA axis regulates blood pressure and vascular tone homeostasis, as well as raises blood glucose levels through gluconeogenesis in the liver during the stress response [37]. Moreover, it is widely known that cortisol signalling in most immune cells generally leads to an immunosuppressive phenotype, which will be discussed later [38].

Both systems convert physical and psychological stressors in the appropriate and situational stress response and are vital for several, if not most, processes in body homeostasis. Dysregulations in these systems may lead to severe disorders, such as a dysfunctional stress response, i.e., stress intolerance. Both the SNS and the HPA axis have been found to be disturbed in several disorders, including chronic pain syndrome [39,40,41,42]. In the following parts, we will discuss the role of both systems in stress intolerance in chronic pain disorders. In addition, we briefly touch upon dysregulations in epigenetic modifications and the immune response, in relation to stress intolerance.

## 5. Sympathetic and Adrenergic Activity Have a Role in Stress Intolerance

Sympathetic dominance as a result of decreased parasympathetic and increased sympathetic activity at baseline has been observed in patients with chronic pain [43,44,45,46,47]. However, the strength of the evidence depends on the clinical aetiology of chronic pain. A meta-analysis by Koenig et al. demonstrated that HRV was consistently decreased only in patients with fibromyalgia and other chronic pain conditions such as pelvic pain, whiplash-associated disorder, and neck-and-shoulder pain [43]. On the contrary, results were conflicting for primary headache or irritable bowel syndrome (IBS) [43]. In addition, the sympathetic stress response in patients with chronic pain is blunted, especially in chronic widespread pain (CWP) syndromes such as fibromyalgia [45,48,49,50,51,52,53]. In other conditions, such as localised chronic muscle pain and chronic whiplash-associated disorder, hypo-reactivity is less pronounced or absent, respectively [48,54].

Biological measures (e.g., catecholamine levels) point in the same direction. On the one hand, noradrenaline levels at baseline have been found to be elevated in patients with fibromyalgia, which is consistent with an increased sympathetic activity [55,56,57,58,59]. On the other hand, changes in noradrenaline and adrenaline in response to different types of stressors are less pronounced, which is consistent with the blunted stress response [58,60,61]. However, results on catecholamine levels in patients with chronic pain remain conflicting as some studies report no or opposite differences at baseline or in response to stress [61,62,63,64].

Autonomic activity has also been associated with various symptoms of stress intolerance [65,66]. Recent systematic reviews concluded that parasympathetic activity was positively associated with self-regulation and pain inhibition capacities, and that cognitive performance is positively associated with HRV [65,66]. Additionally, pain severity showed to be inversely correlated with HRV in an occupational sample comprising people with and without chronic pain. However, this correlation was only significant in the entire sample and in the group without chronic pain, but not in the group with chronic pain, implicating that the autonomic activity of patients with chronic pain relates differently to pain than in those without chronic pain [47]. This is contradicting to the results of Zamunér et al. who demonstrated that pain intensity in fibromyalgia is in fact correlated with sympathetic activity, which is in turn inversely correlated with HRV [67]. Taken together, these results show that sympathetic dominance is associated with symptoms of stress intolerance. Sympathetic dominance might be due to reduced parasympathetic reactivation during recovery from stress, as is the case during recovery from exercise [26].

Preclinical studies also support autonomic involvement in stress intolerance and provide us with deeper insights. Khasar et al. were able to induce hyperalgesia in rats by injection of adrenaline [68]. The hyperalgesia was further enhanced by unexpected sound stress. In addition, removing the adrenal medulla before stress exposure prevented stress-induced enhancement of hyperalgesia [68]. As the adrenal medulla is an important site of adrenaline production, these results indicate that elevated levels of catecholamines are required for the induction of stress-induced hyperalgesia. Their follow-up study later revealed that catecholamines are also pivotal for the maintenance of stress-induced hyperalgesia. Removal of the adrenal medulla after exposure to sound stress reversed the stress-induced hyperalgesia that had occurred in response to stress. Finally, administration of adrenaline in these rats reconstituted the stress-induced hyperalgesia again [69]. These results are in line with another animal study that focussed on the role of α2 ARs, which tightly control noradrenaline release by autoinhibition upon activation. Animals in which the α2 ARs were blocked (through injection of receptor antagonists or knock-out) developed hyperalgesia in response to stress. This stress-induced hyperalgesia was prevented when sympathetic activity was blocked, again showing that sympathetic activity is required for the induction of stress-induced hyperalgesia [70]. Finally, inhibition of the catechol-O-methyltransferase (COMT) enzyme, which prevents the degradation of catecholamines, has been found to increase pain sensitivity through activation of β-ARs [71]. Although some contradictory findings exist [72], accumulating evidence suggests that sympathetic and adrenergic activity may be involved in stress intolerance (see Figure 1).

## 6. The HPA Axis Is Deregulated in Chronic Pain Syndromes

The HPA axis also plays an important role in stress intolerance (see Figure 1 for a summary of findings). Activation of the HPA axis results in an increased concentration of circulating corticosteroids, especially cortisol. Deregulation of adrenal steroid secretion has been reported in several chronic pathological conditions, including chronic stress and dysfunctional chronic pain conditions [73,74]. Alteration of corticosteroid expression can give rise to two opposite phenomena, namely hyper- and hypocortisolism [75].

Hypercortisolism is characterised by basal hypercortisolism and/or hyper-reactivity. Basal hypercortisolism is defined as a permanently increased cortisol level and decreased negative feedback of the HPA axis, whereas hyperreactivity refers to normal cortisol levels with exaggerated behavioural and cortisol responses to stressful events [76]. Hypercortisolism has been reported in several chronic pain conditions, including myofascial pain and burning mouth syndrome [77,78]. Similarly, hypocortisolism includes basal hypocortisolism and hypo-reactivity to stressful events [74]. Tops et al. found that hypocortisolism occurs after a prolonged period of repetitive stimulation of the HPA axis resulting in excessive cortisol release, suggesting that hypocortisolism chronologically follows hypercortisolism [79]. Hypocortisolism has been reported in patients with myalgic encephalomyelitis/chronic fatigue syndrome (ME/CFS), IBS, and chronic pelvic pain [80,81,82]. Interestingly, lower cortisol levels have been associated with lowered pain thresholds and increased pain sensitivity, and a blunted cortisol-awakening response with decreased cognitive function [83,84,85,86]. In CWP and fibromyalgia, contradicting results have been found. Although most findings report hypocortisolism, several studies also reported increased cortisol levels [87,88,89,90,91]. These contradictory results might be partially explained by the fact that the HPA axis can respond differently depending on previous unknown repetitive stressors that have been present in the lives of the participants [92]. One study by Coppens et al., found a blunted cortisol response and a higher subjective stress rating in response to psychological stress in fibromyalgia patients compared to healthy controls [93]. Concerning these inconsistencies, more research is needed to elucidate whether a true causal link between corticosteroid mechanisms and the pathogenesis of chronic pain exists.

Though research has mostly focused on cortisol as measure of the HPA axis function, other components of the axis have also been investigated. CRH is released from the hypothalamus in response to physical and psychological stressors. It interacts with CRH receptors 1 and 2 [94]. CRH exerts actions in both the periphery and stress-related regions in the brain, i.e., the hypothalamus, amygdala, locus coeruleus, and hippocampus. Preclinical research using rat models demonstrated the involvement of CRH in stress-induced hyperalgesia and stress intolerance [95]. Peripheral administration of a CRH receptor 1 antagonist before water avoidance stress inhibited the development of stress-induced visceral hyperalgesia [96,97]. Additionally, in mice exposed to a forced swim test, administration of the CRH receptor 2 antagonist attenuated the development of stress-induced musculoskeletal hyperalgesia [98]. In patients with IBS, administration of the CRH antagonist alpha-helical CRH reduced electrical stimulation-induced abdominal pain [99,100]. Another study found increased pain intensity and decreased pain thresholds as result of rectal distention in healthy volunteers when CRH was peripherally administered [99,100]. Consistent with the preclinical findings, these results strengthen the evidence that CRH and its receptors are involved in stress-induced hyperalgesia and stress intolerance.

## 7. A Key Regulatory Role for Genetics and Epigenetics in Stress Intolerance

Despite accumulating evidence implicating the relevance of the abovementioned systems in stress intolerance in patients with chronic pain, stress responses and pain are variable among and within individuals. For instance, the effect of stress on pain (i.e., hypo- or hyperalgesia in response to stress) depends on the magnitude of the individual stress response [101]. Part of the variability in pain and stress among individuals can be explained by genetics. Genetic polymorphisms affecting the activity of COMT or monoamine oxidase A and B (MAO-A/B), which are both catecholamine-degrading enzymes and thus influence catecholamine levels and ANS functioning, have been associated with increased stress responsiveness and pain sensitivity in both animals and humans [102,103,104,105,106,107,108,109,110,111]. Typically, polymorphisms that lower enzymatic activity and thus elevate catecholamine levels are associated with higher pain sensitivity [112]. Although some conflicting evidence exists [61,62,63,64], these findings are in line with the higher catecholamine levels that have been found in patients with chronic pain.

Genetic polymorphisms of the corticosteroid receptor gene found in chronic pain are also worth mentioning. Macedo et al. found reduced GR expression in combination with the increased prevalence of the MR rs5522 (I180 V) polymorphism in fibromyalgia patients [75]. Other polymorphisms that alter the stress response have also been described. For example, Wüst et al. found that carriers of the GR N363S polymorphism showed increased salivary cortisol response to psychological stimuli, and that the GR *Bcl*I RFLP polymorphism was associated with a diminished cortisol stress response upon psychological stress in healthy individuals [113]. Recently, a study by Linnstaedt et al. found a functional polymorphism in the 3′-UTR of the *FKBP5* gene (rs3800373), a key regulator for glucocorticoid receptor sensitivity, which was associated with a higher chance to develop chronic post-traumatic pain [114]. Finally, the same group found a polymorphism in the corticotropin-releasing hormone binding protein (*CRHBP*) gene (rs7718461) to be highly associated with the *FKBP5* gene, and to be predictive of chronic musculoskeletal pain after a motor vehicle crash [115].

Although genetic polymorphisms can explain at least part of between-subject variability in stress responses and pain [116,117], they cannot explain within-subject variability. Epigenetic changes are strong candidates to explain both variability among individuals and within the same individual as they are dynamic mechanisms, responsive to environmental changes and the context [118]. Only few clinical studies investigated epigenetic changes in relation to chronic pain [119]. The role of epigenetics in the context of stress intolerance in chronic pain has never been investigated in humans, even though epigenetic mechanisms are clearly influenced by acute stress [120,121]. Stress has been reported to influence epigenetic regulation of genes involved in the abovementioned systems. Clinical studies found that DNA methylation—the best-known epigenetic modification—of genes involved in catecholamine degradation (*COMT*, *MAOA,* and *MAOB*) [122,123,124,125] and HPA-axis (*CRHR1, NR3C1*) [126,127,128,129,130] is in fact influenced by early-life stress and altered in patients with stress-related conditions. One study showed that *COMT* DNA methylation associated with lifetime exposure to stress relates to cognitive function in healthy controls [123]. Greater lifetime exposure to stress was associated with reduced *COMT* DNA methylation, which was in turn correlated with reduced working memory accuracy [123]. This study thus supports the involvement of epigenetic mechanisms in stress intolerance as cognitive symptoms, including impaired working memory, may worsen or be triggered in response to stress [13,14].

Genetics and epigenetics are thus both associated with pain and stress. Moreover, genetic polymorphisms can influence DNA methylation in several genes [131,132,133,134,135], as is the case for *COMT* [136,137]. It is thus likely that both genetics and epigenetics underly the role of the ANS and HPA axis in stress intolerance in patients with chronic pain. Of note, the aforementioned studies described stress-related rather than stress-induced epigenetic modifications as all data were obtained from cross-sectional studies. To elucidate a causal and/or regulatory role of epigenetic mechanisms in stress intolerance in chronic pain, future research should investigate the link between acute and chronic stress-induced epigenetic modifications, their downstream effects on the ANS and the HPA axis, and the associated symptoms in both patients with chronic pain and healthy controls.

## 8. Future Directions for Research

Research suggests that patients suffering from chronic pain conditions react differently to stress. However, the biological and physiological mechanisms linking stress and pain remain vague. We introduced the term “stress intolerance”, which refers to the exacerbation or occurrence of symptoms, including but not limited to pain, in response to any type of stress. In this review, we summarised (preliminary) evidence supporting the idea that the two major stress systems, the ANS and the HPA axis, might be able to explain this phenomenon. Furthermore, genetic and epigenetic mechanisms might cover a key regulatory role.

Although evidence indicates that the functionality of the stress systems is deviant in patients with chronic pain, the direction of the link between stress and pain remains unclear. Some studies found that a blunted stress response can predict chronic pain later in life [12,84]. Such results imply that the stress response is already deviant before chronic pain develops. However, other studies could not support this finding [3]. The alternative option is that the stress responses become altered after chronic pain has already developed. This latter option would explain why stress intolerance is common in chronic pain populations. Future studies should thus be designed in a way that would allow us to unravel causal relationships between the two. In a later phase, we can then intervene with the underlying mechanisms and aim to prevent the development of chronic pain and/or the altered stress response.

To date, research on the topic is not only very scarce but the methods and protocols used to measure aspects of the ANS and HPA axis, as well as epigenetic and immune markers, are highly heterogeneous. Consequently, results are often not comparable. Future research methods should be standardised; time of data collection as well as the time between waking up and data collection is crucial and should be clearly reported and standardised. This is especially true when data collection takes place in the morning, due to the cortisol awakening response. We would also suggest employing multiple measurements across several days before and after stress exposure to further control for circadian fluctuations and within-patient variability. Such a design would also allow to investigate the recovery phase after the stressful challenge or event.

Additionally, current research investigated biological outcomes alone, with no link to symptom severity, thus making the available findings less relevant clinically. As stress intolerance is defined by the fluctuations in severity and presence of symptoms after stress exposure, repeated-measure designs investigating solely biological outcomes (without linking them to symptom severity) cannot provide answers on which mechanisms are involved. Future studies should thus also assess symptom severity and biological outcomes at the same time.

Taken together, the current knowledge creates the basis supporting a role for the stress systems in the pathology of chronic pain disorders and specifically stress intolerance. Further studies investigating the stress systems using standardised methods are warranted to obtain a better understanding of the mechanisms at play. A summary of the main future directions for research can be found in Figure 2.

## 9. Future Directions for Clinical Practice

Currently, most physicians provide chronic pain patients with passive and biomedical treatments, which usually consist of medication and surgery. However, this approach often leads to poor benefits and carries a higher risk of adverse events [138]. A biomedical approach to pain omits its multidimensional aspects and disregards the impact of distress, which increases the risk of maintaining the pain experience [139]. Dysfunctional physiological stress response systems add complexity and induce heterogeneity in treatment responses, which emphasises the importance for clinicians of being attentive to stress intolerance.

Several treatment options are available targeting contributing factors to the maintenance of pain and possibly the development of stress intolerance. Educating the patient about pain is relevant in terms of stress management as patients with chronic pain are at higher risk of developing anxiety and depression [139], which in turn have a mediating effect on pain [140,141]. Patient education and reassurance are able to reduce their distress and change their attitudes towards pain [142]. Several systematic reviews with meta-analyses have shown compelling evidence for neuroscience education in reducing pain, perceived disability, and psychosocial factors such as fear-of-movement and catastrophising in patients with chronic pain [142,143,144,145]. Cognitive Behavioural therapy (CBT), acceptance and commitment therapy (ACT), and pain education targeting pain interference, stress, and disability, can also be employed, in an attempt to reduce contributing factors to the pain experience [146,147,148,149].

Sleep is another important contributing factor to chronic pain that should be addressed during management of chronic pain and stress intolerance [150]. The interplay between sleep, stress, and pain has been demonstrated by numerous chronic pain studies, even though the pathophysiology is not fully understood [151,152]. Disrupted sleep results in a low-grade inflammatory response, which will decrease patients’ stress tolerance [153,154]. Clinicians should thus assess sleep problems because sleep deprivation can lead to patients’ inability to face daily stressors [153].

Though the aforementioned approaches have been shown to help reduce pain and increase quality of life, research into the pathophysiological mechanisms of chronic pain and stress intolerance is still much needed. Research into the causal mechanisms may highlight the importance of preventive medicine when results show that the physiological stress response is already deviant before chronic pain develops, as is already shown by some studies [12]. In that case, the development of chronic pain may be prevented by targeting mechanisms underlying a dysregulated stress response. Animal studies already demonstrated that several interventions may be of help in targeting a dysregulated stress response. Both physical activity and antidepressant administration have been found to attenuate stress-induced DNA-methylation changes in rats [155,156]. By understanding the effect of various interventions on stress-induced epigenetic changes, we might be able to target key dysregulations underlying stress intolerance.

## Figures and Tables

**Figure 1 jcm-12-02245-f001:**
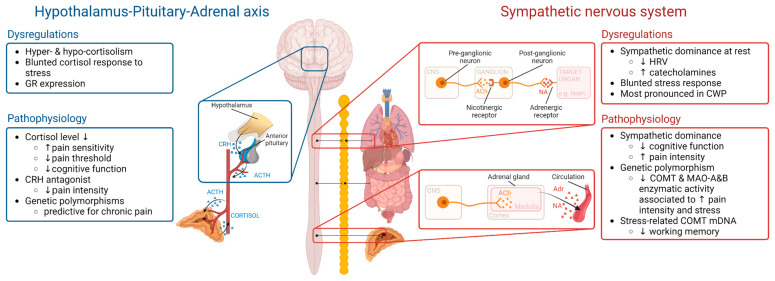
Visual representation of the major stress pathways, together with common dysregulations in chronic pain and their possible pathophysiological implications in stress intolerance. ↑, Increased; ↓, Decreased; ACh, Acetylcholine; Adr, Adrenaline; ACTH, Adrenocorticotropic hormone; COMT, Catechol-O-methyltransferase; CNS, Central nervous system; CWP, Chronic widespread pain; CRH, Corticotropin-releasing hormone; mDNA, DNA methylation; GR, Glucocorticoid receptor; HRV, Heart rate variability; HPA axis, Hypothalamic-pituitary-adrenal axis; MAO-A&B, Monoamine oxidase A&B; NA, Noradrenaline. Created with BioRender.com (Accessed on 9 February 2023).

**Figure 2 jcm-12-02245-f002:**
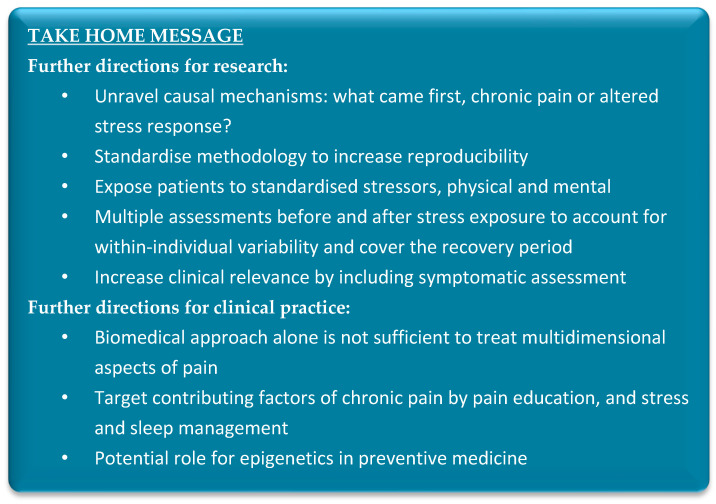
Summary of future directions for research and clinical practice.

## Data Availability

Not applicable.

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
