# Peer review of "The Biology of Stress Intolerance in Patients with Chronic Pain—State of the Art and Future Directions"

_jcm, 2023, doi:10.3390/jcm12062245_

Round 1

Reviewer 1 Report

General and specific comments to JCM-2242114 entitled „The biology of stress intolerance in patients with chronic pain – state of the art and future directions“ (Wyns et al.).

This is an interesting narrative review on the contribution of stress on chronic pain and vice versa.

Several points should be considered additionally.

1.     Line 100 and 104: The statements about the influence of AR activation on HRV are misleading. A low HRV is associated with alpha AR stimulation and a high HRV with beta HRV stimulation.

2.     The authors mentioned the contradictory findings regarding basal cortisol levels in chronic pain (see, e.g., lines 215-217). However, a major reason for this seemingly contradictory finding could be the concept of repetitive stressors that could end up leading to dysfunction of the HPA axis in terms of a lack of adequate response to stressors. This concept (of allostatic load) was introduced most notably by McEwan (NEJM 1998;338(3):171-179) and has been developed since. This should be discussed and elaborated throughout the manuscript.

3.     The key components of stress mechanisms should be mentioned, especially the link between the endocannabinoid system (ECS) and the HPA axis (Morena M et al, Neuropsychopharmacology 2016;41(1):80-102). The ECS is also involved in the development of chronic pain. Therefore, dysfunction of the ECS could be an important link between chronic stress and chronic pain. This should be discussed throughout the manuscript.

4.     Basically, at the neurobiological level, it is assumed that chronic pain represents a chronic inflammatory state. Against this background, it is important to mention that chronic stress also has an impact on the immune system. This is all the more relevant as only recently significant findings indicate that fibromyalgia may be a functional autoimmune disease (Goebel A et al, J Clin Invest 2021;131(13):e144201). This implies that both immune system and nociceptive system dysfunction are adversely intertwined. Please draw attention to this.

5.     It has been hypothesized that genetic and epigenetic susceptibility could, to some extent, explain the development of chronic pain (see, e.g., lines 266-275). In this regard, an in-depth analysis, at least at an exploratory level, was performed in a recent publication (Buhck M et al. J Affect Disord 2022;308:466-472).

6.     It was also mentioned that there are no "dose" response data for the effects of pain due to chronic stress (see, e.g., lines 335ff). This is not entirely correct. Several studies of post-traumatic stress disorder (PTSD) showed a dose-response effect between the extent of trauma and clinical outcomes (Forbes D et al, J Clin Psychiatry 2013;75(2):147-153; Akerblom S et al, Clin J Pain 2018;34(6):487-496). Interestingly, trauma and PTSD are massively associated with chronic pain (Akerblom S et al, Clin J Pain 2018;34(6):487-496). This has been shown down to the epigenetic level (Achenbach J et al, Clin Epigenetics 2019;11(1):126). These findings should be mentioned.

7.     Line 214: It should be „blunted”

8.     Line 325: Delete the first “is”.

Reviewer 2 Report

The manuscript reviews (state-of-the-art and future directions) the potential role of biological stress intolerance in patients with chronic pain. This is a relevant topic for the JCM and the authors have done a good job. Specifically, the role of the autonomic nervous system (ANS) and hypothalamus-pituitary-adrenal (HPA) axis is summarized. Inconsistencies in evidence and future challenges for research and clinical practice are described.

The main strength of the study is the breadth and depth of the review. I only have some minor concerns for the authors:

- Although this is not a systematic review since data from the search is provided, it could be expanded by reporting details such as the flowchart or if it was carried out by peers.

- Consider revising some cites and references on patient education and psychological interventions:

* p3 l359-360: this reference does not seem appropriate since the study is not on patient education.

*p3 l363: this reference seems to be better for CBT than for neuroscience education.

p3 l363-366: references 140-142 are confusing to me. They are cited as examples of CBT but they are not. Reference 140 and 142 is a meta-analysis of ACT and mindfulness-based therapies. Reference 141 is on MBSR. Consider revising this paragraph and its references.

Round 2

Reviewer 1 Report

The manuscript has greatly improved. I have no further comments.